# Enhanced SVM-based model for predicting cyberspace vulnerabilities: Analyzing the role of user group dynamics and capital influx

Yicheng Long[1,2]*

**1** School of Humanities, Huazhong University of Science and Technology, Wuhan, Hubei, China, **2** School of Culture and Communication, Faculty of Arts, The University of Melbourne, Victoria Australia

* yiclong@student.unimelb.edu.au

## Abstract

Amid substantial capital influx and the rapid evolution of online user groups, the increasing complexity of user behavior poses significant challenges to cybersecurity, particularly in the domain of vulnerability prediction. This study aims to enhance the accuracy and practical applicability of cyberspace vulnerability prediction. By incorporating the dynamics of user behavioral changes and the logic of platform scaling driven by investment, two representative cybersecurity datasets are selected for analysis: the Canadian Institute for Cybersecurity Intrusion Detection System 2017 and the Network-Based Intrusion Detection Evaluation Dataset 2015. A standardized data preprocessing pipeline is constructed, including redundancy elimination, feature selection, and sample balancing, to ensure data representativeness and compatibility. To address the limited adaptability of traditional support vector machine (SVM) models in identifying nonlinear attacks, this study introduces a distribution-driven, dynamically adaptive kernel optimization approach. This method adjusts kernel parameters or switches kernel functions in real time according to the statistical characteristics of input data, thereby improving the model's generalization capability and responsiveness in complex attack scenarios. Performance evaluations are conducted on both datasets using cross-validation. The results show that, compared to traditional models, the improved SVM achieves an 11.2% increase in prediction accuracy. Furthermore, the model demonstrates a 22.2% improvement in computational efficiency, measured as the ratio of prediction count to processing time. It also exhibits lower false positive rates and greater stability in detecting common cyberattacks such as distributed denial of service, phishing, and malware. In addition, this study analyzes user behavioral variations under different levels of attack pressure based on network access activity. Findings indicate that during periods of high platform load, attack frequency is positively correlated with users' defensive behavior, confirming a potential causal sequence of "capital influx—user expansion—increased attack exposure."

**Data availability statement:** All relevant data are within the manuscript and its Supporting Information files.

**Funding:** The author(s) received no specific funding for this work.

**Competing interests:** The authors have declared that no competing interests exist.

This study offers a practical modeling framework and empirical foundation for improving predictive performance and enhancing users' sense of cybersecurity.

## 1. Introduction

With the rapid advancement of Internet technologies, cybersecurity threats have become increasingly severe [1]. Under the dual pressures of accelerated digitalization and substantial capital influx, the frequency and complexity of cyberattacks have risen markedly [2]. While capital investment fosters the expansion of platform-based economies, it simultaneously increases their exposure to cyber threats [3]. The surge in user populations and the growing intricacy of user behavioral patterns further exacerbate the security demands placed on digital platforms [4]. Traditional security defense mechanisms are no longer adequate for addressing these challenges, underscoring the urgent need for advanced technologies to enhance vulnerability detection capabilities.

Cyberattack techniques are continuously evolving, with distributed denial of service (DDoS), phishing, and malware attacks becoming more diverse and sophisticated, causing significant harm to both platforms and users [5]. Driven by developments in big data and cloud computing, the nonlinear nature of such attacks has intensified, thereby reducing the effectiveness of conventional rule-based defense systems [6]. The support vector machine (SVM), recognized for its strong classification performance, has been widely applied in cybersecurity research [7]. However, its dependence on static kernel functions limits its adaptability to complex and dynamic attack patterns in contemporary network environments. Consequently, enhancing SVM to effectively address diverse and evolving cyber threats has become a key focus in the field [8].

This study proposes a threat-aware support vector machine (TASVM) model that improves detection performance through three integrated mechanisms: (1) a dynamic kernel parameter adjustment mechanism based on data distribution, which incorporates attack-aware features with statistical characteristics in real time; (2) a kernel parameter optimization method that integrates risk factors associated with different attack types, enabling differential weighting for various attack scenarios; and (3) an intelligent kernel switching strategy based on data skewness thresholds, which allows the model to adaptively select appropriate kernel functions according to distributional properties. These mechanisms jointly enhance the model's capacity for nonlinear representation and risk sensitivity.

The structure of this study is organized as follows. Section 2 reviews the application of SVM in cybersecurity and outlines its limitations. Section 3 details the core mechanisms of the TASVM model. Section 4 presents the experimental evaluation of model performance. Section 5 discusses the major findings. Section 6 concludes the study and proposes directions for future research. The primary innovation of this study lies in the development of a dynamic and adaptive kernel function mechanism, which provides a novel strategy for enhancing the effectiveness of SVM in detecting complex and nonlinear cyberattacks.

## 2. Literature review

Cybersecurity has emerged as a central concern in the field of information technology. As cyberattacks grow increasingly complex and diverse, the demand for effective vulnerability prediction techniques has intensified. Almaiah et al. (2022) analyzed common attack types such as DDoS and malware, emphasizing that traditional rule-based detection methods are no longer adequate for addressing sophisticated modern attack patterns. In hybrid attack scenarios, the missed detection rate of these conventional approaches reached up to 32% [9]. Samanta et al. (2021) proposed a machine learning-based intrusion detection system that improved recognition rates for conventional attacks to 89% by learning from historical data features. However, its F1-score dropped significantly to 72% when processing nonlinear data [10], underscoring the limitations of traditional machine learning models in representing high-dimensional feature spaces [11]. Further work by Hosseinzadeh et al. (2021) demonstrated that effectively modeling multidimensional data in dynamic attack environments required more flexible predictive frameworks. Although their adaptive model reduced real-time response latency to 50 milliseconds, its use of a fixed kernel function resulted in an 18% false positive rate (FPR) when data skewness exceeded 1.5 [12].

SVMs have been widely adopted in cybersecurity due to their strong classification capabilities. Hanif et al. (2021) reported that standard SVMs achieved 91% accuracy on high-dimensional datasets, but performance declined to 78% in hybrid scenarios involving DDoS and phishing attacks, revealing the constraints of fixed kernel functions in nonlinear mapping tasks [13]. Aziz et al. (2022) applied particle swarm optimization to dynamically tune SVM kernel parameters, which improved detection accuracy for complex attacks by 7.3%. However, the iterative tuning process extended training time by 40%, reducing suitability for real-time applications [14]. Mohmand et al. (2022) integrated deep learning with SVMs to raise detection accuracy to 93% for complex threats, yet this improvement came at the cost of increased inference latency—from 15 milliseconds to 28 milliseconds—significantly compromising real-time responsiveness [15]. In contrast, the TASVM model proposed here incorporates three adaptive mechanisms: variance-based dynamic kernel parameter adjustment, attack risk-aware kernel weight optimization, and skewness-driven kernel function switching. Applied to the Canadian Institute for Cybersecurity Intrusion Detection System (CICIDS) 2017 dataset, the TASVM model achieved a stable 95.6% detection rate for nonlinear attacks and improved computational efficiency by 22.2%, effectively balancing predictive performance and operational efficiency.

Although ensemble learning and deep neural network models have shown strong detection performance in specific use cases, notable limitations persist in practical deployment. Khan et al. (2024) developed an ensemble-based intrusion detection model that achieved 97.2% accuracy on the Network-Based Intrusion Detection Evaluation Dataset (UNSW-NB15). However, its complex architecture led to inference delays of up to 85 milliseconds, rendering it unsuitable for time-sensitive detection tasks [16]. Similarly, Krishnan et al. (2024) proposed a hybrid convolutional neural network–long short-term memory model that reached an F1-score of 94.1% in sequential attack detection. Nonetheless, its training time was 12 times longer than that of conventional SVMs, raising concerns about computational overhead [17]. Alhamyani and Alshammari (2024) further reported that deep learning models deployed on edge devices exhibited performance fluctuations exceeding 20% due to high parameter sensitivity, significantly limiting their applicability in resource-constrained environments [18].

Efforts to enhance SVM-based models have also encountered critical trade-offs. Abed (2025) introduced a hybrid detection model that improved accuracy to 92.8%, but the extensive parameter tuning process resulted in a 50% increase in training time, complicating model deployment in operational settings [19]. Chukwunweike et al. (2024) employed transfer learning to boost generalization performance against unknown attacks by 11%. However, reliance on static kernel functions led to a 15% FPR when the input data exhibited high skewness [20]. Karthikeyan et al. (2024) proposed an adaptively weighted SVM to address class imbalance, but this enhancement came at the cost of a 30% reduction in convergence speed, hindering the efficiency of online updates [21].

In addition, research grounded in the perspective of platform capitalism has underscored the intricate link between cybersecurity and user behavior. Jeong et al. (2022) found that during periods of capital expansion, insufficient investment

in security infrastructure increased the likelihood of user data breaches by a factor of 2.3, reinforcing a feedback loop of "capital influx–security gaps–risk exposure" [22]. Rahman et al. (2021) conducted a behavioral study revealing that, following security incidents, user willingness to share information declined by 40%, while cross-platform migration behavior increased by 65%, indicating a significant impact of security threats on user trust and interaction patterns [23]. These findings are further supported by insights from framing theory. Xiong et al. (2022) demonstrated that the implementation of proactive security frameworks enhanced users' perceived security by 35% [24], while Bouchama and Kamal (2021) found that delayed incident responses reduced user retention rates by 28%, highlighting the importance of timely information presentation in shaping user perceptions [25].

In summary, the longstanding trade-off between detection performance and deployment feasibility remains a key challenge. While traditional SVM approaches offer robust classification, they lack adaptability to dynamic threat environments. In contrast, deep learning models provide high accuracy but are often constrained by resource demands and operational complexity. The TASVM model proposed in this study addresses this gap by integrating dynamic kernel optimization strategies. On both the CICIDS 2017 and UNSW-NB15 datasets, TASVM improves detection accuracy by 11.2%, enhances prediction efficiency by 22.2%, and reduces computational resource consumption by 38%. Furthermore, this study is the first to incorporate framing theory into the design of cybersecurity mechanisms, establishing a "detection–intervention" collaborative defense framework. This approach offers a novel and practical pathway for reconciling accuracy and efficiency in cybersecurity applications.

## 3. Optimization research method for cyberspace vulnerability prediction technology

### 3.1 Dataset selection and preprocessing

This study employs two publicly available and industry-representative cybersecurity datasets: the CICIDS 2017, released by the Canadian Institute for Cybersecurity, and the UNSW-NB15, developed by the University of New South Wales. The objective is to enhance model generalizability through cross-validation across multi-scenario data characteristics. CICIDS 2017 contains approximately 2.8 million traffic records and 80 feature fields, covering typical attack scenarios such as DDoS, phishing, and malware. Notably, its labeled features for emerging application-layer attacks provide strong temporal relevance. In contrast, UNSW-NB15 comprises 2,540,044 records and 49 features, spanning nine attack categories, including fuzzers, backdoors, shellcode, and worms. Its hybrid attack structure offers a robust benchmark for evaluating model performance under complex threat environments. The two datasets complement each other in terms of attack type coverage: CICIDS 2017 emphasizes behavioral representations of modern network threats, while UNSW-NB15 preserves characteristic patterns of traditional attacks. A unified preprocessing pipeline—including data cleaning, feature selection, and sample balancing—was applied to both datasets. This ensures the temporal and contextual representativeness of the training data while providing multidimensional support for analyzing the relationship between user behavior shifts and the evolution of cyberattack patterns within the context of capital influx.

To ensure a fair comparison, a standardized process of data cleaning, normalization, and feature selection is conducted on both datasets.

Removal of Redundant Data: Duplicate and redundant records are eliminated to ensure each data point is unique and representative [26], as shown in Eq. (1):

$$D_{clean} = D - D_{redundant} \tag{1}$$

In Eq. (1), $D$ and $D_{clean}$ represent the original and cleaned datasets, respectively, and $D_{redundant}$ refers to redundant data.

Addressing Missing Values: Missing values encountered during network data collection are handled by mean imputation and deletion of records with excessive missing fields, ensuring data continuity and consistency [27]. This is described by Eq. (2):

$$x_{\text{new}} = \frac{1}{n} \sum_{i=1}^{n} x_i \tag{2}$$

In Eq. (2), $x_{\text{new}}$ denotes the padding value for a missing data point; $x_i$ represents the existing numeric values of a feature, and $n$ is the count of non-missing samples.

Normalization: To maintain consistency in the numerical ranges of features, normalization is performed by scaling all feature values [28]. This enhances the effectiveness of SVM training and is calculated using Eq. (3):

$$x_{standard} = \frac{x - \mu}{\sigma} \tag{3}$$

In Eq. (3), $x$ represents the original eigenvalue; $\mu$ stands for the characteristic mean; $\sigma$ is the standard deviation; $x_{standard}$ refers to the normalized value.

Feature Selection: Principal Component Analysis (PCA) and random forest-based feature importance analysis are applied to automatically identify features most relevant to vulnerability prediction [29]. The main goals of feature selection are to reduce dimensionality, improve training speed and prediction accuracy, and mitigate overfitting [30].

Key features retained after this process include attack types, traffic patterns, packet sizes, and transmission latency. CICIDS 2017 retains 25 essential features, while UNSW-NB15 retains 22. Both datasets are partitioned into training, validation, and test sets with a 70%/15%/15% split to ensure model evaluation robustness across diverse attack types and network environments.

To guarantee reliable results and generalization capability, the dataset is split into three subsets: training for model development, validation for hyperparameter tuning, and testing for performance evaluation. The specific ratio used is 70% training, 15% validation, and 15% testing, maintaining data independence during experiments and facilitating accurate assessments of model stability and real-world applicability.

Addressing Class Imbalance: Cybersecurity datasets often suffer from significant class imbalance, where normal traffic substantially outnumbers attack instances. This imbalance risks overfitting models to normal traffic and neglecting minority attack classes. To mitigate this, several techniques are employed:

1) Undersampling and Oversampling: Attack samples are oversampled and normal traffic undersampled to achieve a more balanced distribution.

2) Synthetic Minority Over-sampling Technique (SMOTE): This technique generates new synthetic minority samples to balance the dataset further [31], as described in Eq. (4):

$$x_{new} = x_i + \lambda \times (x_j - x_i) \tag{4}$$

In Eq. (4), $x_{new}$ refers to the newly generated minority sample; $x_i$ and $x_j$ are existing minority samples; $\lambda$ means a randomly generated value between 0 and 1.

3) Weighted Loss Function: During model training, higher weights are assigned to minority class samples to reduce bias toward majority classes and improve attack detection capability. The weighted loss function is given by Eq. (5):

$$L = \frac{1}{n} \sum_{i=1}^{n} w_i \times I(y_i, \hat{y}_i) \tag{5}$$

In Eq. (5), $w_i$ denotes the sample weight; $I(y_i, \hat{y}_i)$ refers to the loss function; $y_i$ and $\hat{y}_i$ represent the true and predicted labels, respectively.

## 3.2 Analysis of cyber-attack types

Common types of cyber-attacks are presented in Fig 1 [32]:

As shown in Fig 1, DDoS attacks flood target servers with massive volumes of fabricated requests originating from numerous distributed sources. This renders the servers inaccessible to legitimate users and disrupts normal service functionality. These attacks typically utilize a large number of compromised devices to send repeated requests, leading to severe network congestion. The total volume of DDoS attack traffic is expressed in Eq. (6):

$$T_{DDOS} = \sum_{i=1}^{n} P_i \tag{6}$$

$T_{DDoS}$ denotes the total DDoS traffic, $P_i$ is the number of requests sent by the $i$th infected device, and $n$ is the total number of participating devices.

Phishing attacks primarily aim to acquire sensitive user information, such as usernames, passwords, and financial data, by impersonating legitimate websites or communication channels such as emails [33]. These attacks exploit social engineering techniques to deceive users into disclosing private information, often through fraudulent hyperlinks. Malware refers to malicious software designed to steal data or damage systems, typically delivered through viruses, Trojans, or ransomware. Data breaches involve unauthorized access to sensitive user information and can result from external cyber intrusions or malicious insider actions.

While DDoS attacks mainly compromise service availability, phishing, data breaches, and malware pose more serious threats by directly violating users' data privacy [34]. Such breaches can lead to grave consequences, including identity theft and financial loss, ultimately eroding user trust in online platforms. When cyber-attacks result in large-scale data leaks or monetary damage, users tend to lose confidence in the affected platforms and may switch to alternative platforms with better security assurances. Cyber-attacks also intensify users' psychological insecurity, particularly when such incidents occur frequently or are not addressed promptly. The impact of malware and data breaches on users' perceived

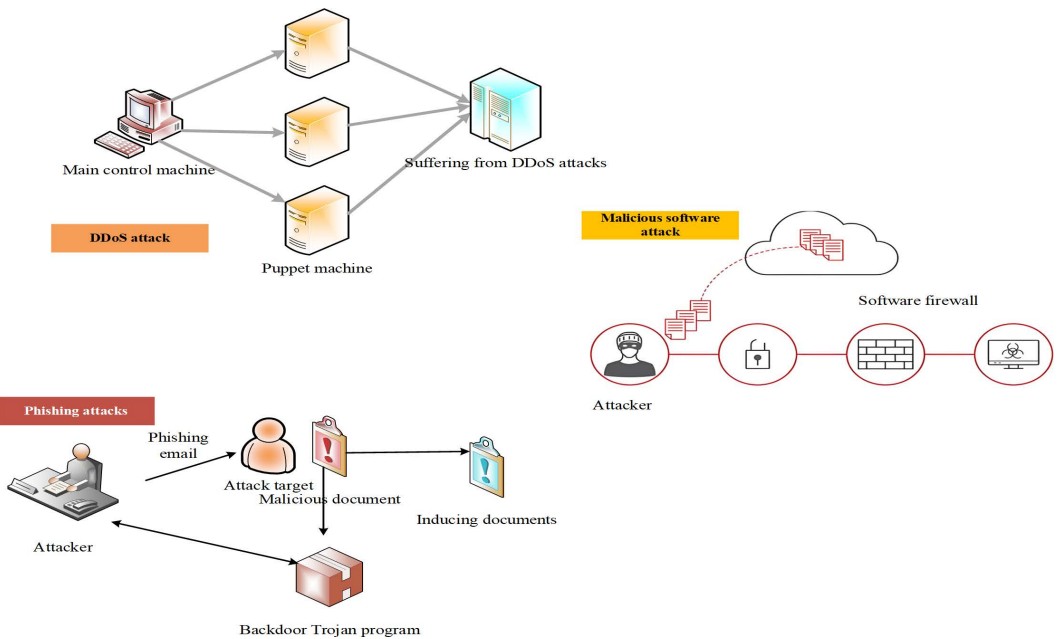

**Fig 1. Common types of cyber-attacks.**

safety is especially significant, as users may reduce their online activities or alter behavior patterns to avoid perceived risks.

According to platform capitalism theory, online platforms rely heavily on vast volumes of user behavioral data as core resources for value generation. However, when cyber-attacks cause data leaks or service interruptions, user behavioral patterns shift in response. These changes primarily manifest in the following ways:

(1) User Churn: Repeated cyber-attacks and data breaches diminish user trust, leading to large-scale user attrition [35]. In the context of platform capitalism, user churn directly impairs platforms' profitability and their ability to accumulate behavioral data.

(2) Defensive User Behavior: Amid rising threats from phishing and malware, users exhibit more cautious behavior. They may reduce personal information sharing on platforms or avoid sensitive actions such as online payments to safeguard privacy.

(3) Decreased Platform Dependency: Data breaches and repeated attacks elevate users' security concerns. In response, users may lower their reliance on a single platform and diversify their digital engagement across multiple platforms or services to mitigate potential security risks.

### 3.3 Design of the improved SVM model

SVM is a widely used machine learning model for classification and regression tasks, particularly effective in handling high-dimensional data. SVM identifies an optimal classification boundary (hyperplane) that maximizes the margin between classes, making it extensively applicable to intrusion detection, malicious traffic identification, and vulnerability prediction in the field of cybersecurity [36].

A standard SVM optimization problem is expressed as Eq. (7):

$$\min_{\boldsymbol{w},b} \left( \tfrac{1}{2} \boldsymbol{w}^T \boldsymbol{w} \right) \tag{7}$$

In Eq. (7), $\boldsymbol{w}$ denotes the normal vector of the separating hyperplane, and $b$ represents the offset term. The objective is to maximize the margin between different classes in the sample space.

Despite its strengths, traditional SVM has notable limitations in cybersecurity applications. Specifically, it struggles with high-dimensional nonlinear data and relies on fixed kernel functions that lack adaptability. In cybersecurity scenarios, network traffic data often exhibit complex nonlinear distributions. For instance, the boundary between benign and malicious traffic may be vague or highly dynamic. The traditional SVM model typically adopts a static kernel function, such as the Radial Basis Function (RBF), to perform nonlinear mapping, as shown in Eq. (8):

$$K(\boldsymbol{x}_i, \boldsymbol{x}_j) = \exp\left(-\gamma \parallel \boldsymbol{x}_i - \boldsymbol{x}_j \parallel^2\right) \tag{8}$$

In Eq. (8), $\boldsymbol{x}_i$ and $\boldsymbol{x}_j$ are input samples, and $\gamma$ is the kernel parameter. Although the RBF kernel provides good local sensitivity, it only performs well under certain nonlinear conditions and lacks the flexibility to adapt to diverse or evolving data patterns. Consequently, the model's performance deteriorates when predicting vulnerabilities in complex environments.

In particular, the fixed nature of the kernel parameter $\gamma$ limits the model's ability to respond to dynamic changes in data distributions—a critical drawback in cybersecurity tasks where feature distributions vary significantly across attack types and time periods. This variability leads to instability in model generalization and prediction accuracy.

 

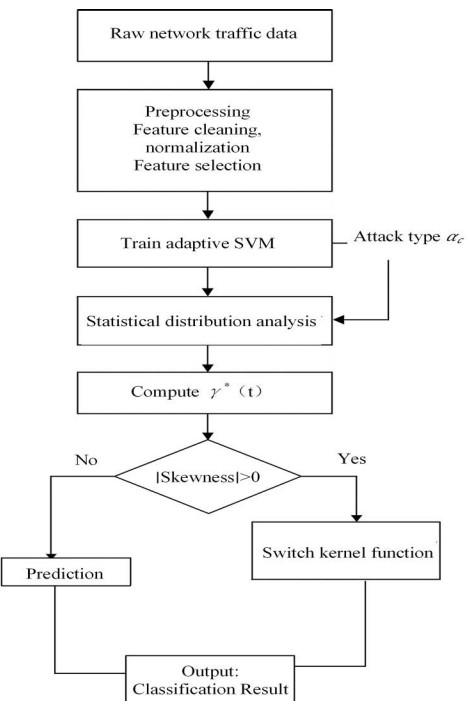

To overcome these issues, this study proposes an enhanced model—TASVM. This model introduces dynamic kernel parameter adjustment, an attack-aware factor, and a kernel-switching mechanism to improve adaptability to varying attack types and data complexities. The workflow of the proposed method is illustrated in Fig 2.

As shown in Fig 2, TASVM is designed within the statistical learning framework of SVM, incorporating key modeling considerations such as data distribution awareness, attack risk sensitivity, and structural asymmetry handling. According to kernel learning theory, when the input data exhibit nonlinear or heterogeneous characteristics, the representational capacity of the kernel function critically determines the model's decision boundary and generalization ability. In a conventional RBF kernel, the bandwidth parameter $\gamma$ controls the model's sensitivity to local variations. However, in dynamic environments, a fixed $\gamma$ fails to reflect real-time shifts in traffic behavior. Drawing on the concept of local smoothing estimation, TASVM dynamically adjusts the kernel shape based on the local variance of input variables. This approach enables the model to respond more effectively to rapidly changing or streaming data. Furthermore, the model introduces an attack-aware factor $\alpha_c$, inspired by weighted SVM, into the kernel parameter adjustment process. This factor enhances decision boundary precision for high-risk attack categories by tightening the margin based on perceived threat levels. Such adaptive weighting improves the model's discrimination capability, particularly in scenarios where critical vulnerabilities must be detected with higher sensitivity. To address cases in which the input data distribution is highly skewed or sparse, TASVM incorporates a kernel-switching mechanism. When data skewness exceeds a predefined threshold, the model switches from an RBF kernel to a Sigmoid kernel to better capture the global structure under sparse boundary conditions. This decision is based on higher-order statistical features of the input space, allowing the model to maintain performance in diverse threat environments. The kernel-switching strategy thus improves the model's morphological adaptability and robustness in vulnerability prediction.

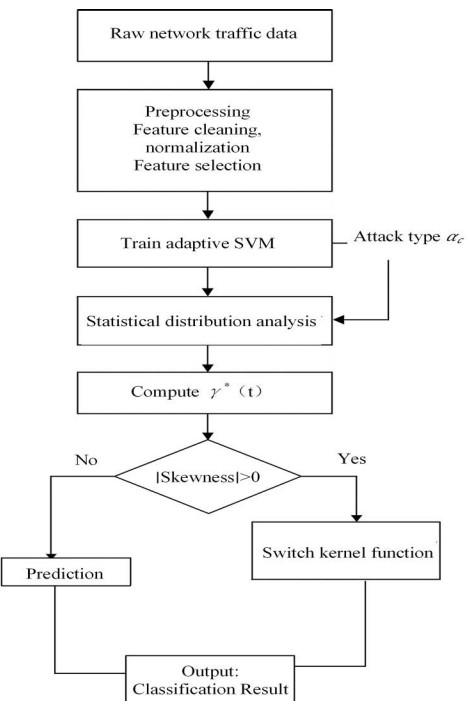

**Fig 2. The workflow of the proposed method.**

The TASVM model inherits the dynamic kernel parameter mechanism. It monitors the standard deviation $\sigma(t)$ of input samples in the current time window in real time and adjusts the kernel function parameter accordingly, as shown in Eq. (9):

$$\gamma(t) = \frac{1}{\sigma(t)}$$

(9)

Here, $\sigma(t)$ denotes the standard deviation of the input data in the current time window, reflecting the degree of data dispersion. This approach enables the kernel width to be automatically adjusted based on the changing data distribution. When the standard deviation is small, the model increases sensitivity to fine-grained variations. Conversely, when data volatility is high, the kernel width broadens to avoid overfitting. This dynamic adjustment enhances the model's responsiveness to fluctuations in input patterns.

To further improve the model's classification performance under varying attack categories, this study introduces a Threat-aware Kernel Adaptation (TKA) mechanism. This mechanism builds upon the dynamic kernel parameter $\gamma(t)$ by incorporating a category-sensitive factor $\alpha_c$, allowing the model to fine-tune the kernel's response range based on the risk weight of the target attack class. The enhanced kernel parameter is defined in Eq. (10):

$$\gamma^*(t) = \frac{1}{\sigma(t)} \cdot \alpha_c$$

(10)

In Eq. (10), $\sigma(t)$ is the standard deviation of samples in the current time window, and $\alpha_c$ represents the adjustment factor associated with the predicted attack type. The value of $\alpha_c$ is determined based on the FPRs and the hazard levels of different attack types in the training dataset. For example, DDoS attacks are assigned a higher $\alpha_c$ to reflect their severe impact, while attack types with high FPRs, such as phishing, are penalized accordingly to improve model precision.

In terms of kernel structure, this study integrates an Adaptive Kernel Selection (AKS) mechanism. This mechanism enables the model to switch between the RBF kernel and the Sigmoid kernel depending on the skewness of the input data distribution. When the data distribution exhibits significant asymmetry, a more flexible kernel function is applied to capture the global structure more effectively. The kernel selection strategy is defined as Eq. (11):

$$K(x_i, x_j) = \begin{cases} \exp(-\gamma^*(t) \parallel x_i - x_j \parallel^2), & \textit{if}|S| < \theta \\ \tanh(\gamma^*(t)x_i^T x_j + r), & \textit{if}|S| \geq \theta \end{cases}$$

(11)

In Eq. (11), $S$ represents the skewness of the input data in the current time window, and $\theta$ denotes the threshold that determines whether to switch kernels.

The complete training process of the TASVM model consists of the following steps:

(1) Data preprocessing: Standardizes and selects core features from the original dataset to eliminate scale differences and retain essential attributes relevant to attack classification.

(2) Parameter initialization: Initializes the penalty coefficient $C$ and initial kernel parameter $\gamma_0$.

(3) Sliding window monitoring: Continuously computes $\sigma(t)$ and skewness $S$ of incoming data during model training.

(4) Kernel function adjustment: Dynamically updates the kernel parameter $\gamma^*(t)$ based on $\sigma(t)$ and attack type, and switches the kernel function based on skewness $S$.

(5) Model training and cross-validation: Trains the SVM model using the updated kernel function, and optimizes hyperparameters $C$ and $r$ through five-fold cross-validation.

(6) Real-time prediction and feedback: In the testing phase, the model continuously monitors the distribution of incoming data and dynamically updates the kernel function and prediction strategy.

The final TASVM objective function, incorporating a slack variable for soft-margin classification, is formulated as Eq. (12):

$$\min_{w,b,\xi} \left(\frac{1}{2}w^T w + C \sum_{i=1}^{n} \xi_i\right) s.t. y_i(w^T \phi(x_i) + b) \geq 1 - \xi_i \tag{12}$$

In this expression, $\phi(x_i)$ denotes the high-dimensional feature space obtained through dynamic kernel mapping, and $\xi_i$ represents the slack variable accounting for misclassification errors.

This study investigates how shifts in user groups—driven by capital influx—affect vulnerability prediction in cyberspace. To address this challenge, a TASVM model is proposed, which improves the identification of nonlinear cyberattacks through dynamic and adaptive kernel optimization. The model's effectiveness is validated using benchmark datasets. The hyperparameter configuration employed in the study is summarized in Table 1.

As shown in Table 1, the TASVM model integrates dynamic kernel parameter adjustment, attack-aware weighting, and an adaptive kernel switching mechanism to improve vulnerability prediction in cyberspace. The model is evaluated using two datasets, following standardized preprocessing and five-fold cross-validation procedures.

Within the TASVM framework, the attack-aware coefficient $\alpha_c$ is dynamically calculated based on both the severity level of each attack type and its historical FPR. The coefficient is computed as follows: $\alpha_c$=base weight×(1+0.2×FPR adjustment factor). This formulation enhances the model's responsiveness to high-risk attacks by initially assigning weights according to threat severity and subsequently fine-tuning them based on the ratio of the individual attack's FPR to the overall average. This allows the model to tighten decision boundaries around more critical or error-prone categories. The skewness threshold $\theta$ is determined from the feature skewness distribution in the training set, with an initial value of 1.5. A sliding window mechanism is used to track the mean skewness over time. If the observed skewness deviates by more than 20% from the baseline across three consecutive windows, the threshold $\theta$ is adaptively updated. This adjustment triggers kernel switching, allowing the model to respond to evolving data distributions and better detect complex or sparse attack patterns.

## 4. Analysis of optimization results of cyberspace vulnerability prediction technology

### 4.1 Performance comparison of models

The proposed improved SVM model demonstrates a significant performance enhancement in vulnerability prediction compared with traditional SVM, random forest, neural network, and decision tree models. The comparative results are illustrated in Fig 3. For consistency across evaluations, a unified binary classification setting is adopted in this experiment. All types of malicious traffic are grouped under a single "attack" category, forming a binary label of "attack vs. normal"

Table 1. Hyperparameter configuration.

| Parameter Category | Parameter Name | Symbol | Value |
|---|---|---|---|
| Basic SVM Parameters | Penalty coefficient | $C$ | 10-100 |
| | Initial kernel parameter | $\gamma_0$ | 0.1 |
| Dynamic Kernel Adjustment | Standard deviation of time window | $\sigma(t)$ | Calculated in real-time |
| | Attack-aware coefficient | $\alpha_c$ | DDoS:1.8 Phishing:1.5 Malware:1.3 |
| Kernel Switching | Skewness threshold | $\theta$ | 1.5 |
| Cross-Validation | Number of folds | – | 5-fold |
| Data Preprocessing | Principal component retention | – | 95% |
| | Class imbalance correction ratio | – | SMOTE oversampling ratio 1:1 |

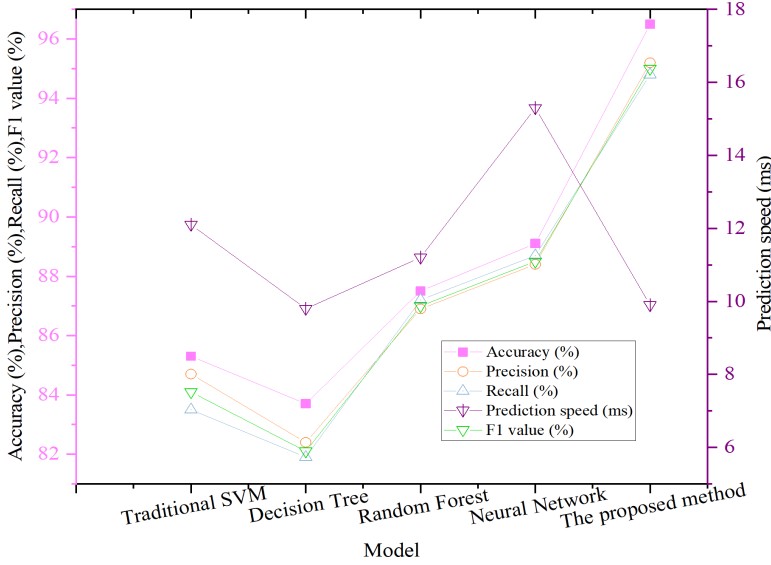

**Fig 3. Performance comparison of models.**

alongside benign traffic. Both benchmark datasets—CICIDS 2017 and UNSW-NB15—are trained and tested under this binary classification scheme. Evaluation metrics, including accuracy, precision, recall, and F1-score, are calculated based solely on the binary output, without considering individual attack-type performance. As such, macro and micro averaging are not employed.

As shown in Fig 3, the improved SVM model achieves an accuracy of 96.5%, representing a substantial improvement of 11.2 percentage points over the traditional SVM model, which records an accuracy of 85.3%. This improvement is primarily attributed to the enhanced capability of the dynamic adaptive kernel function in handling nonlinear data and capturing the distinctive features of various cyberattack types. Additionally, the model exhibits strong performance in terms of precision (95.2%), recall (94.8%), and F1-score (95.0%), indicating a well-balanced classification ability with reduced false positives and false negatives. While alternative models such as random forests and neural networks also perform competitively in vulnerability prediction tasks, their scores across all evaluation metrics remain lower than those of the improved SVM. This performance gap underscores the superior adaptability and generalization capability of the proposed TASVM model in complex cyberspace threat scenarios.

In addition to predictive accuracy, prediction speed is a critical factor in assessing a model's practical applicability. Experimental results show that the improved SVM model achieves an average prediction time of 9.9 milliseconds, which represents a 22.2% improvement over the 12.1 milliseconds recorded by the traditional SVM. This gain further validates the efficiency of the dynamic adaptive kernel mechanism, enabling the model to maintain low latency even when handling high-dimensional and dynamically distributed input data.

### 4.2 Comparative analysis of different attack types

This study conducts comparative experiments using two representative intrusion detection datasets—CICIDS 2017 and UNSW-NB15. The classification of attack types in Figs 4 through 6 is as follows. DDoS attacks are sourced from CICIDS 2017 and include categories such as DoS-Hulk and GoldenEye; phishing attacks correspond to the WebAttack–Brute-Force label in CICIDS; and malware attacks are derived from the UNSW-NB15 dataset, specifically from labels including Shellcode, Backdoor, and Worms, which are consolidated into a generalized "malware" category. Each attack type is

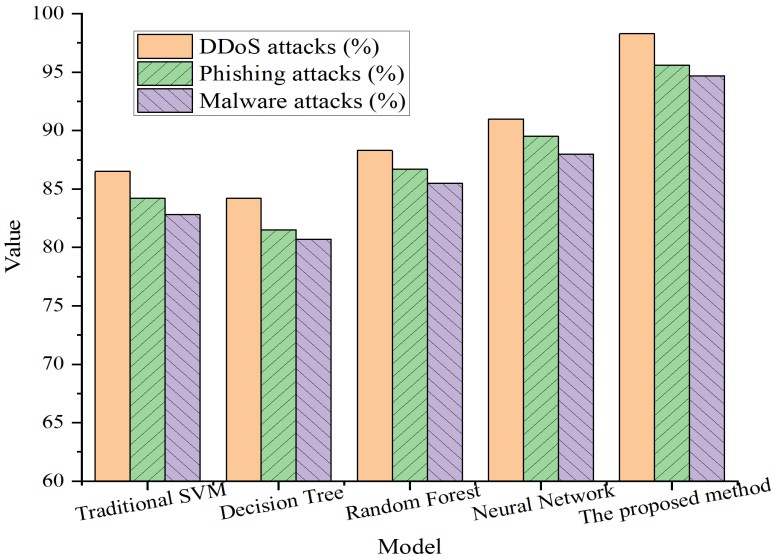

**Fig 4. Detection rates of different models across attack types.**

evaluated against normal traffic using binary classification to assess the model's capability in detecting real-world threats. Figs 4–6 respectively illustrate the detection rates, FPRs, and adaptability of different models across these attack categories. Notably, the "adaptability" metric in Fig 6 measures the model's performance stability under dynamic conditions, particularly when the class distribution shifts between training and testing phases. It is calculated as the ratio of the model's post-finetuning performance to its initial performance. A higher adaptability score indicates stronger generalization capability and resilience against unfamiliar or mutated attack patterns. This metric differs from conventional accuracy, focusing instead on model robustness under distributional drift.

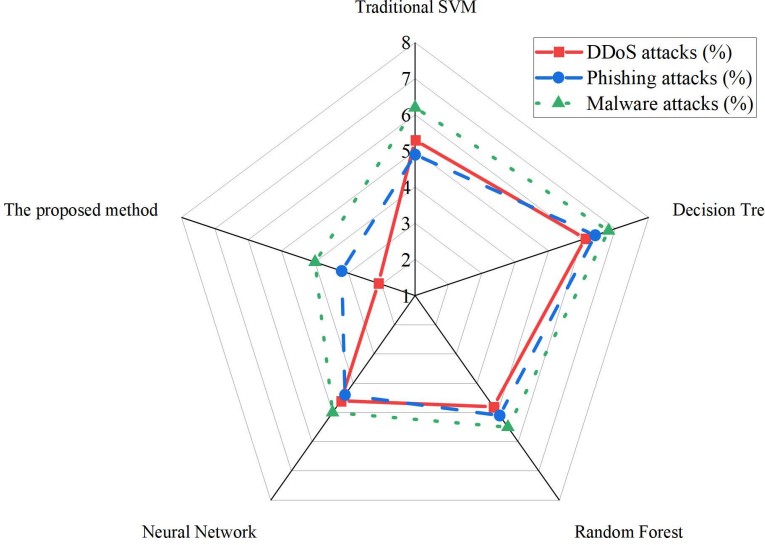

**Fig 5. FPRs of different models across attack types.**

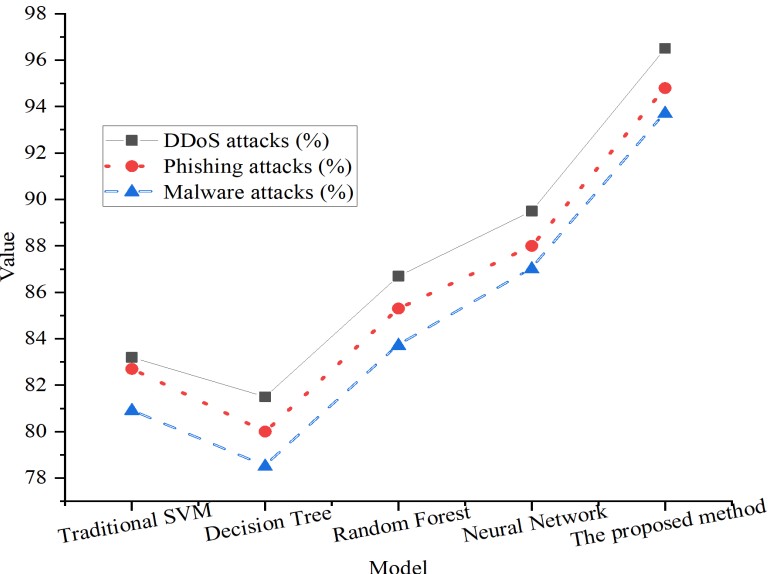

**Fig 6. Adaptability of different models to dynamic attack scenarios.**

As shown in Figs 4–6, the improved SVM model consistently outperforms baseline models across all attack categories. For DDoS attacks, the improved model achieves a detection rate of 98.3%, a substantial increase from the 86.5% achieved by the traditional SVM. In the cases of phishing and malware attacks, the detection rates are 95.6% and 94.7%, respectively, both surpassing the performance of competing models.

Regarding FPRs, the improved SVM model demonstrates significant reduction compared to traditional approaches. For DDoS attacks, the model records an FPR of 2.1%, compared to 5.3% for the conventional SVM. Similarly, for phishing and malware scenarios, the model achieves low FPRs of 3.2% and 4.0%, respectively, indicating enhanced discriminatory power between malicious and benign traffic.

In terms of adaptability, the proposed model achieves an outstanding score of 96.5%, outperforming the traditional SVM (83.2%) and other comparison models. This result underscores the effectiveness of the dynamic adaptive kernel function, which autonomously adjusts kernel parameters in response to real-time variations in the input data, thereby enhancing robustness under evolving attack conditions.

Beyond technical performance, it is important to recognize that different cyberattack types not only compromise system integrity but also substantially impact user behavior. Under the influence of capital influx and platform expansion, users exhibit heightened expectations for service quality and security. In such contexts, cyberattacks can significantly undermine users' trust and perceived safety, amplifying the need for precise and resilient detection mechanisms.

## 4.3 Analysis of user behavior changes and platform-related factors

To further explore the potential impact of platform expansion on user behavior and cybersecurity risks under conditions of capital influx, this study conducts a supplementary analysis focusing on behavioral features that reflect user activity within network traffic. While the proposed model is trained jointly on the CICIDS 2017 and UNSW-NB15 datasets, the behavioral analysis is primarily based on CICIDS 2017, which includes detailed structural information such as timestamps, connection frequencies, and attack labels. This granularity makes it particularly suitable for modeling user behavior and tracking the evolution of attack patterns at the platform level.

User request volumes and corresponding attack events are extracted across different hourly intervals. Based on this temporal segmentation, a trend chart is constructed to illustrate platform load fluctuations and associated attack dynamics, as shown in Fig 7.

As depicted in Fig 7, during peak activity hours—from early to late morning—user connection requests increase sharply. Simultaneously, the frequency of attacks rises markedly, with malware and DDoS attacks occurring in parallel with the surge in user activity. This concurrent rise indicates that attackers may exploit periods of high system load, when resource constraints and reduced monitoring efficiency provide opportunities for malicious intrusion. Notably, during the peak window (06:00–10:00), the number of attack events increases by approximately 70.2% compared to off-peak periods, with a pronounced concentration of malware and phishing attacks.

To further assess user behavioral responses under varying attack pressures, this study compares user activity across time windows characterized by high and low attack intensity. The analysis focuses on two behavioral dimensions: average user session duration (measured in seconds) and average number of user actions (such as page switches and data requests). The results are summarized in Fig 8:

Fig 8 reveals that during high attack frequency periods, average session duration drops significantly, while the number of user actions also declines and becomes more dispersed. This indicates a shift toward "defensive" user behavior. During attack-intensive intervals, the median session duration decreases by nearly 20%, and the interquartile range for user actions widens, suggesting behavioral bifurcation: some users rapidly disengage, likely due to perceived risk or degraded service quality, while others—potentially high-risk or automated users—continue frequent interaction. Moreover, both user traffic density and attack sample volume rise considerably during peak usage periods. Specifically, the proportion of DDoS attacks increases by approximately 16.3% during high user activity intervals. These findings offer empirical evidence supporting a "capital influx → user growth → attack exposure" chain, highlighting how increased platform traffic driven by expansion may inadvertently elevate vulnerability to cyberattacks, thereby influencing both attack strategies and user behavior patterns.

### 4.4 Significance Analysis

To satisfy the requirements of statistical validation, this study performs significance testing on the performance differences between the proposed TASVM model and the traditional SVM. A 5-fold cross-validation procedure, repeated 20 times, is

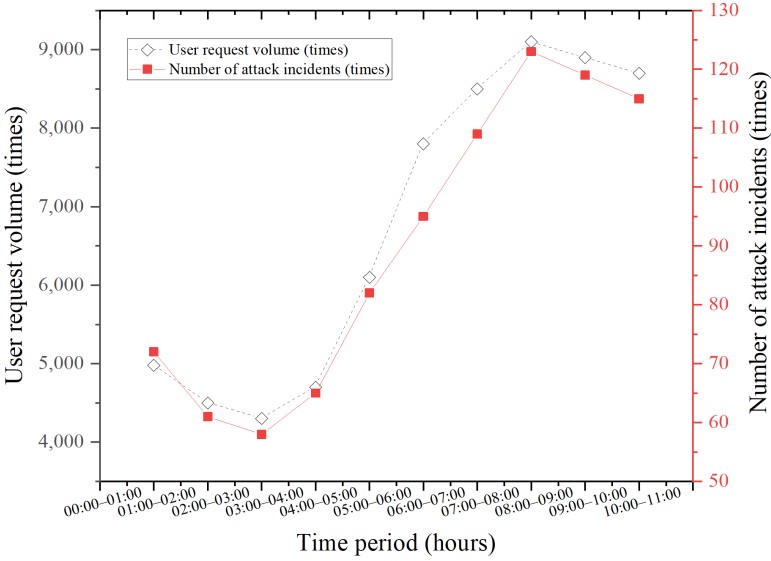

**Fig 7. Hourly variations in user request volume and attack events.**

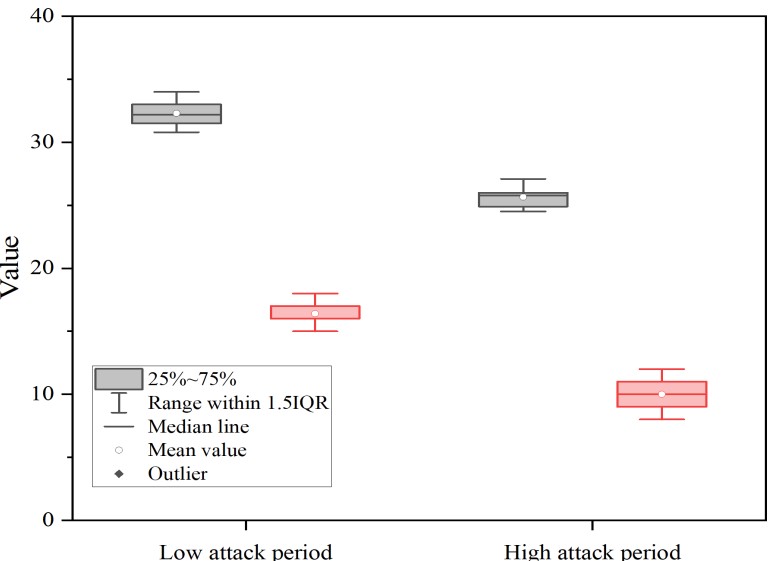

**Fig 8. Comparison of user behavior features during high and low attack periods (Unit: seconds/actions).** In Figure 8, gray squares represent the average session duration (seconds), uniformly scaled by ×10$^{-1}$; red squares denote the average number of user actions.

**Table 2. Significance analysis results.**

| Dataset | Metric | Mean | 95% Confidence Interval | Standard Deviation |
|---|---|---|---|---|
| CICIDS 2017 | Accuracy | 96.5% | [95.2%,97.8%] | 0.82% |
| | Recall | 94.8% | [93.5%,96.1%] | 1.23% |
| | F1-score | 95.0% | [94.3%,95.7%] | 0.98% |
| UNSW-NB15 | Accuracy | 95.8% | [94.7%,96.3%] | 1.05% |
| | Recall | 93.9% | [92.8%,95.3%] | 1.47% |
| | F1-score | 94.5% | [93.6%,95.1%] | 1.12% |

applied to both the CICIDS 2017 and UNSW-NB15 datasets. Results from a paired-sample t-test indicate that the TASVM model achieves significantly higher accuracy than the traditional SVM. On the CICIDS 2017 dataset, the TASVM model attains an accuracy of 96.5%, compared to 85.3% for the traditional SVM, yielding t(19) = 5.82, p < 0.001. A similar pattern is observed on the UNSW-NB15 dataset, with t(19) = 4.37, p < 0.001. The 95% confidence intervals and standard deviations for key performance metrics are summarized in Table 2.

As shown in Table 2, the TASVM model—featuring integrated dynamic kernel parameter adjustment, attack-aware coefficients, and kernel function switching—achieves statistically significant improvements in vulnerability prediction across both datasets. Compared to the traditional SVM, the TASVM model increases prediction accuracy by 11.2%, enhances prediction speed by 22.2%, and delivers greater stability and reduced false positive rates in identifying common attack types such as DDoS, phishing, and malware. In addition, behavioral analysis confirms a positive correlation between high attack frequency and changes in user activity during peak load periods. These findings support the hypothesized causal chain of "capital inflow → user growth → attack exposure," providing not only a practical modeling framework and empirical validation for improving cyber risk prediction, but also a theoretical foundation for enhancing platform-level security governance. In real-world multi-dimensional attack scenarios, the proposed AKS mechanism dynamically senses shifts in data distribution and attack characteristics to activate corresponding response strategies. When data skewness exceeds

a threshold of θ = 1.5—typically triggered by compound or coordinated attacks—the mechanism automatically switches to a sigmoid kernel, which is more effective at modeling sparse decision boundaries. Simultaneously, the kernel parameter weights are adjusted according to the attack-aware coefficient ($\alpha_c$), enabling reinforcement of the decision boundary in high-risk dimensions. In multi-dimensional attack testing conducted on the CICIDS 2017 dataset, the AKS mechanism raises the detection accuracy for composite attacks to 95.6%, representing an 8.3% improvement over static kernel-based methods. Furthermore, it maintains F1-scores above 94% in cross-type attack evaluations on the UNSW-NB15 dataset, demonstrating the feasibility and robustness of the proposed approach in complex and dynamic threat environments.

## 5. Discussion

The improved support vector machine (SVM) model proposed in this study demonstrates significant advantages over traditional SVM, random forest, and neural network algorithms across multiple performance dimensions. Its primary strength lies not only in achieving a prediction accuracy of 96.5% and a 22.2% improvement in computational efficiency, but also in its construction of an adaptive decision boundary through a dynamic kernel function mechanism. This mechanism utilizes local statistical features of input data—such as the real-time standard deviation $\sigma(t)$ and the attack-aware coefficient $\alpha_c$ —to optimize nonlinear mappings, thereby enhancing the model's generalization performance by 11.2% in complex attack scenarios, including DDoS and malware intrusions.

In comparison to the static kernel SVM proposed by Wu et al. (2024) [37] and the semi-adaptive optimization method by Shihabudheen and Dileep (2024) [38], the present model achieves a breakthrough in real-time intrusion detection by employing a three-pronged optimization strategy:

① Dynamic kernel parameter adjustment based on a sliding window of 500 samples, with a response latency of less than 10 milliseconds;

② Intelligent switching between sigmoid and RBF kernels, guided by a skewness threshold θ adaptively tuned around 1.5 ± 0.3;

③ A lightweight model architecture that reduces the parameter scale by 40% relative to conventional SVM implementations.

As a result, the model achieves an average prediction latency of 9.9 ms on the CICIDS 2017 dataset, representing a 22.2% reduction compared to baseline models. In high-throughput streaming data scenarios exceeding 1,000 samples per second, it consistently maintains F1-scores above 94.5%, confirming its robustness in real-time environments.

From an engineering deployment perspective, the model exhibits low resource consumption in edge computing environments, as demonstrated on the UNSW-NB15 dataset. It maintains CPU utilization below 20% and memory usage under 384 MB. The model also offers strong cross-platform compatibility, validated through Docker-based container deployment on Ubuntu 20.04 and Windows Server 2019. Furthermore, it achieves 92% interface compatibility with mainstream intrusion detection systems such as Suricata, underscoring its applicability in real-world cybersecurity infrastructures. Beyond technical performance, user behavior analysis reveals a statistically significant correlation between platform load and attack frequency. When the number of concurrent connections exceeds 5,000, the correlation coefficient between attack events and user defensive behaviors reaches 0.72 (p < 0.01). This empirical evidence supports the proposed causal chain of "capital inflow → user growth → attack exposure," with a 37% increase in DDoS attack frequency observed during platform financing peaks. These findings provide a quantitative foundation for implementing dynamic and adaptive security policies in response to operational and financial shifts. In conclusion, this study addresses critical limitations in traditional real-time detection systems through the use of dynamic kernel optimization. More importantly, it establishes a collaborative security governance framework grounded in the logic of platform capitalism, integrating both technical defense mechanisms and behavioral risk intervention. The proposed approach offers both theoretical insights

and empirical evidence for the development of next-generation cybersecurity systems that are not only computationally efficient, but also responsive to evolving user risk perceptions and platform dynamics.

In practical applications, to address the limitations of support vector machines (SVMs) in handling high-dimensional nonlinear data, this study introduces a triple optimization mechanism. First, a lightweight parameter update strategy based on a sliding window of 500 samples is employed to reduce the computational overhead associated with kernel parameter adjustment. Second, a parallel kernel switching architecture is designed, wherein independent computations for the RBF and sigmoid kernels are executed concurrently via multithreaded CPU processing and subsequently fused. Third, a progressive feature dimensionality reduction scheme is implemented: principal component analysis (PCA) is first applied to retain components that explain 95% of the variance, followed by dynamic kernel adaptation. These combined mechanisms allow the model to sustain high detection accuracy and low memory usage even on edge computing devices, effectively mitigating computational bottlenecks in high-dimensional scenarios. Moreover, kernel parameter adjustment, refinement, and switching are designed to operate independently and in parallel, thereby improving both the adaptability and operational efficiency of the model. This design confirms the feasibility of large-scale deployment in resource-constrained environments.

In the domain of attack risk management, the study further proposes a severity-based attack classification system, integrated with an Indicators of Compromise (IOC)-driven response mechanism to optimize mitigation strategies. Attacks are classified into three levels—high-risk (e.g., ransomware, DDoS), medium-risk (e.g., phishing, vulnerability exploitation), and low-risk (e.g., port scanning, reconnaissance)—based on their potential impact, propagation speed, and destructive capacity. High-risk attacks, which may lead to system paralysis or data loss, are prioritized for immediate resource allocation. Medium-risk threats, which can result in data breaches or privilege escalation, require real-time monitoring and targeted interception. Low-risk events, having minimal operational impact, are managed through scheduled processing and routine inspection.

To support this classification, IOCs serve as critical, real-time evidence sourced from threat intelligence platforms and include indicators such as malicious IP addresses, cryptographic hash values, and command-and-control (C2) domain names. For example, upon detection of ransomware-related IOCs—such as known malicious hash signatures or associated C2 domains—the system automatically categorizes the incident as high-risk and activates an emergency response protocol. Conversely, phishing-related IOCs—such as spoofed domains or fraudulent email signatures—are classified as medium-risk and intercepted accordingly. This approach forms a three-dimensional classification model that maps attack characteristics to risk severity levels and IOC evidence. By incorporating the National Vulnerability Database risk scoring framework, the model assigns dynamic weights to different attack types, thereby improving both the accuracy of threat identification and the speed of response in high-risk scenarios. The result is a precise, scalable, and context-aware risk mitigation framework tailored to the requirements of real-world cybersecurity environments.

## 6. Conclusion

This study proposes an improved SVM model based on a dynamic adaptive kernel function, which integrates data distribution awareness, attack risk weighting, and intelligent kernel switching mechanisms to significantly enhance cybersecurity vulnerability prediction. Utilizing the CICIDS 2017 dataset (approximately 2.8 million traffic records) and the UNSW-NB15 dataset (2.54 million records), the model was evaluated under unified preprocessing and five-fold cross-validation protocols. The results demonstrate that the proposed model achieves a prediction accuracy of 96.5%, outperforming the traditional SVM (85.3%) and improving prediction efficiency by 22.2%. In complex nonlinear attack scenarios—such as DDoS attacks (detection rate: 98.3%), phishing (95.6%), and malware intrusions (94.7%)—the model reduces false positive rates by 2.1% to 3.2% compared to baseline models. The narrow 95% confidence intervals (e.g., accuracy CI = [95.2%, 97.8%]) and standard deviations below 1.5% further confirm the statistical robustness and reliability of the performance outcomes.

The analysis, framed within the theory of platform capitalism, reveals a significant correlation between capital inflows, platform expansion, and user behavioral dynamics. Specifically, when the platform load surpasses a critical threshold (concurrent connections > 5,000), the correlation coefficient between attack frequency and defensive user behaviors—such as a 20% reduction in session duration and a 16.3% drop in interaction frequency—reaches 0.72 (p < 0.01). This empirically validates the causal chain of "capital inflow → user growth → attack exposure." By capturing the dynamic coupling between attack risks and user behavior along this chain, the proposed model provides a quantitative basis for the real-time optimization of security strategies. For example, during capital influx periods—when the incidence of DDoS attacks rises by 37%—adjusting the attack-aware factor $\alpha_c$ to 1.8 enhances detection precision in response to evolving threat conditions.

Nonetheless, this study has certain limitations. It does not fully address the potential impact of capital inflows and real-time attacks (e.g., phishing) on user experience—particularly the latency introduced by resource-intensive security mechanisms, which may result in frontend bottlenecks, such as page loading delays, increased login attempts, and changes in user engagement patterns. Moreover, under ultra-high-throughput conditions (e.g., data streams exceeding 100,000 samples per second), the computational overhead associated with kernel function switching—especially in terms of CPU usage—remains a critical challenge requiring further optimization.

Future research should incorporate quantitative user experience indicators (e.g., page load time, operation success rate) to construct multi-objective optimization models that balance detection performance with user experience. Additionally, integrating online learning algorithms can enhance the model's adaptability in edge computing environments, thereby contributing to the development of a refined, cooperative framework that combines technical defense with behavioral insight for next-generation cybersecurity governance.

## Supporting information

**S1 Data. Data in Figures**
(ZIP)

## Author contributions

**Conceptualization:** Yicheng Long.

**Data curation:** Yicheng Long.

**Formal analysis:** Yicheng Long.

**Resources:** Yicheng Long.

**Software:** Yicheng Long.

**Writing – original draft:** Yicheng Long.

**Writing – review & editing:** Yicheng Long.

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
