## [Decision Letter · Decision Letter 0]

PONE-D-25-20493Application of the Improved Support Vector Machine Model for Cyberspace Vulnerability Prediction Technology from the Perspective of User Group Changes Based on Capital InfluxPLOS ONE

Dear Dr. Long,

Thank you for submitting your manuscript to PLOS ONE. After careful consideration, we feel that it has merit but does not fully meet PLOS ONE’s publication criteria as it currently stands. Therefore, we invite you to submit a revised version of the manuscript that addresses the points raised during the review process.

We look forward to receiving your revised manuscript.

Kind regards,

Pankaj Bhambri, Ph.D.

Academic Editor

PLOS ONE

Journal Requirements:

Additional Editor Comments :

Minor Revisions are Required.

Reviewers' comments:

Reviewer's Responses to Questions

**Comments to the Author**

1. Is the manuscript technically sound, and do the data support the conclusions?

Reviewer #1: Partly

Reviewer #2: Yes

Reviewer #3: Yes

Reviewer #4: Yes

2. Has the statistical analysis been performed appropriately and rigorously? 

Reviewer #1: No

Reviewer #2: Yes

Reviewer #3: Yes

Reviewer #4: Yes

3. Have the authors made all data underlying the findings in their manuscript fully available?

Reviewer #1: Yes

Reviewer #2: Yes

Reviewer #3: Yes

Reviewer #4: Yes

4. Is the manuscript presented in an intelligible fashion and written in standard English?

Reviewer #1: No

Reviewer #2: Yes

Reviewer #3: Yes

Reviewer #4: Yes

5. Review Comments to the Author

Reviewer #1: The manuscript titled "Application of the Improved Support Vector Machine Model for Cyberspace Vulnerability Prediction Technology from the Perspective of User Group Changes Based on Capital Influx" presents a novel approach using a Threat-Aware Support Vector Machine (TASVM) model with adaptive kernel optimization to improve vulnerability prediction in cyberspace. The paper addresses a relevant and timely problem and demonstrates meaningful performance improvements over traditional SVM, decision tree, neural network, and random forest models.

However, several important concerns limit the manuscript’s readiness for publication in its current form:

1. Technical Soundness-

The proposed TASVM model integrates multiple dynamic kernel mechanisms to enhance adaptability to nonlinear attacks, which is a promising contribution. The use of CICIDS 2017 and UNSW-NB15 datasets is appropriate, and the experimental results are aligned with the study’s goals. However, the lack of detail in implementation (e.g., hyperparameter tuning strategies, exact kernel switching conditions, and code reproducibility) and no mention of model limitations in scalability and computation under real-time conditions reduce confidence in the system’s practical viability.

2. Statistical Analysis-

The manuscript does not include significance testing, confidence intervals, or variability measures (e.g., standard deviation across cross-validation folds). While average accuracy and other performance metrics are reported, the absence of statistical validation undermines the strength of the conclusions. This is a critical shortcoming that must be addressed to meet PLOS ONE's standards.

3. Language and Clarity-

Although the manuscript is generally intelligible, the writing includes numerous grammatical issues, awkward sentence constructions, and stylistic inconsistencies. These issues negatively impact readability and should be corrected through professional language editing.

4. Data Availability-

The authors appropriately use two public datasets and state that all data are available within the manuscript and supporting information, which meets the journal’s data availability requirement.

5. Additional Suggestions-

Include significance testing (e.g., t-tests or ANOVA) to statistically validate model improvements. Add variability metrics (e.g., standard deviation, confidence intervals) for reported performance scores. Clarify how the attack-aware factor (αc) and skewness threshold (θ) are calculated and adjusted. Provide runtime benchmarks or discuss deployment feasibility for real-time systems. Improve figures for clarity, and ensure all abbreviations are defined on first use. Submit the manuscript for professional language editing.

Reviewer #2: Thank you to the Author and Editor(s) for this opportunity to provide feedback.

This Original Submission is well-written in both standard and scientific English, well-organized, and its data are largely unproblematic; its original contribution to scholarly research is also evident. I do not have any concerns about its equations or statistical explanations.

In the real world, organized cyberattacks are rarely categoric or unidimensional, often integrating approaches by disdaining loss, following the path of least resistance and concentrating efforts. I wonder if the Author thinks it is pertinent to include a more detailed comparison of how the AKS mechanism is mobilized against multidimensional attacks, since this would be the best representation of feasibility and the project is designed in-itself for performance.

In real-world applications, the SVM property of “insufficient processing capability of high-dimensional nonlinear data” poses a problem for advanced computation requirements. How does the Author propose to circumvent clear obstacles to implementation? For example, would parameter adjustment, parameter refinement and kernel function switching be feasible simultaneous yet separate processes at larger scales for consumers? I propose the Author include some text about implementation potential, especially since past reviewers seemed primarily skeptical due to an inherent supposition of novel computation leading to restricted implementation.

In terms of capital influx some form of concomitance or proposed simultaneity is certain, being that cyberthreat acting which is not directly strategic in terms of impact (DDoS, Ransomware) is nevertheless perpetrated in real-time (Phishing). However, Figure 7 likely also represents user response to critical frontend impacts of cyberthreat intrusion or intrusion detection since resource-intensive data collection, response and prevention processes as well as backend measures drive exponential bottleneck changes to user experience causing a greater number of attempted logins, varying user response to load times, etc. Should some mention of this be made in the limitations, in the form of identifying that trade-offs may be certain?

Reviewer #3: 1. In order to improve clarity and focus, consider a revised version title like “Enhanced SVM-Based Model for Predicting Cyberspace Vulnerabilities: Analyzing the Role of User Group Dynamics and Capital Influx”

2. Some sentences are dense and hard to follow (e.g., “by integrating the dynamics of user behavior change with the logic of platform capital expansion”). Try to simplify language and use more direct phrasing without diluting the academic tone.

3. Terms like “prediction efficiency” and “platform capital expansion” may be vague or open to multiple interpretations. Thus, replace these with more conventional terms like computational efficiency or platform scaling due to investment.

4. Eliminate redundancy, especially in the description of TASVM's innovation. Avoid restating the same contributions multiple times.

Reviewer #4: - Comparative analysis of literature review is missing need to add along with their results readings

- In the discussion and conclusion section, no numerical results reading was discussed.

- when dealing with attack risk not seen any category of attack basis on severity or risk of attack to deal woth it on priority.

- indicator of compromise role in the attack detection and risk of attack is totally neglected in the said study, can be a major factor and need to discussed and find from true positive attacks.

6. PLOS authors have the option to publish the peer review history of their article (what does this mean? ). If published, this will include your full peer review and any attached files.

**Do you want your identity to be public for this peer review?** For information about this choice, including consent withdrawal, please see our Privacy Policy .

Reviewer #1: No

Reviewer #2: **Yes: ** Paul-André Betito, HBA*, MSW, RSW

Reviewer #3: No

Reviewer #4: No

---

## [Author Response · Author response to Decision Letter 1]

11 Jun 2025

PONE-D-25-20493

Application of the Improved Support Vector Machine Model for Cyberspace Vulnerability Prediction Technology from the Perspective of User Group Changes Based on Capital Influx

PLOS ONE

Dear Dr. Long,

Thank you for submitting your manuscript to PLOS ONE. After careful consideration, we feel that it has merit but does not fully meet PLOS ONE’s publication criteria as it currently stands. Therefore, we invite you to submit a revised version of the manuscript that addresses the points raised during the review process.

We look forward to receiving your revised manuscript.

Kind regards,

Pankaj Bhambri, Ph.D.

Academic Editor

PLOS ONE

Journal Requirements:

Additional Editor Comments :

Minor Revisions are Required.

Reviewers' comments:

Reviewer's Responses to Questions

Comments to the Author

1. Is the manuscript technically sound, and do the data support the conclusions?

Reviewer #1: Partly

Reviewer #2: Yes

Reviewer #3: Yes

Reviewer #4: Yes

2. Has the statistical analysis been performed appropriately and rigorously?

Reviewer #1: No

Reviewer #2: Yes

Reviewer #3: Yes

Reviewer #4: Yes

3. Have the authors made all data underlying the findings in their manuscript fully available?

Reviewer #1: Yes

Reviewer #2: Yes

Reviewer #3: Yes

Reviewer #4: Yes

4. Is the manuscript presented in an intelligible fashion and written in standard English?

Reviewer #1: No

Reviewer #2: Yes

Reviewer #3: Yes

Reviewer #4: Yes

5. Review Comments to the Author

Reviewer #1: The manuscript titled "Application of the Improved Support Vector Machine Model for Cyberspace Vulnerability Prediction Technology from the Perspective of User Group Changes Based on Capital Influx" presents a novel approach using a Threat-Aware Support Vector Machine (TASVM) model with adaptive kernel optimization to improve vulnerability prediction in cyberspace. The paper addresses a relevant and timely problem and demonstrates meaningful performance improvements over traditional SVM, decision tree, neural network, and random forest models.

However, several important concerns limit the manuscript’s readiness for publication in its current form:

1. Technical Soundness-

The proposed TASVM model integrates multiple dynamic kernel mechanisms to enhance adaptability to nonlinear attacks, which is a promising contribution. The use of CICIDS 2017 and UNSW-NB15 datasets is appropriate, and the experimental results are aligned with the study’s goals. However, the lack of detail in implementation (e.g., hyperparameter tuning strategies, exact kernel switching conditions, and code reproducibility) and no mention of model limitations in scalability and computation under real-time conditions reduce confidence in the system’s practical viability.

Reply: Thank you for your comments. According to your suggestions, the content of Section 3.3 has been revised to discuss the experimental implementation details of this paper, including providing the design results of model hyperparameters and discussing the scalability and computational limitations of the model under real-time conditions. As shown in Table 1, the threat-aware support vector machine model proposed in this study integrates dynamic kernel parameter adjustment, attack perception factors, and kernel switching mechanisms for cyberspace vulnerability prediction. Two datasets were selected, and after data preprocessing, the model performance was tested through cross-validation.

2. Statistical Analysis-

The manuscript does not include significance testing, confidence intervals, or variability measures (e.g., standard deviation across cross-validation folds). While average accuracy and other performance metrics are reported, the absence of statistical validation undermines the strength of the conclusions. This is a critical shortcoming that must be addressed to meet PLOS ONE's standards.

Reply: Thank you for your suggestions. According to your advice, the content of the results section in this paper has been revised, including adding the results of significance analysis and conducting an in-depth discussion to improve the quality of this paper. As shown in Table 2, the threat-aware support vector machine model proposed in this study, by integrating dynamic kernel parameter adjustment, attack perception factors, and kernel function switching mechanisms, has achieved significant improvements in cyberspace vulnerability prediction performance on the CICIDS 2017 and UNSW-NB15 datasets.

3. Language and Clarity-

Although the manuscript is generally intelligible, the writing includes numerous grammatical issues, awkward sentence constructions, and stylistic inconsistencies. These issues negatively impact readability and should be corrected through professional language editing.

Reply: Thank you for your reminder. In accordance with your reminder, the entire content of this paper has been comprehensively proofread, and the grammar and sentences have been fully optimized to comprehensively improve the readability and content quality of this paper.

4. Data Availability-

The authors appropriately use two public datasets and state that all data are available within the manuscript and supporting information, which meets the journal’s data availability requirement.

Reply: Thank you for your comments. Thank you very much for reviewing and guiding this paper. Based on your suggestions, this paper has been revised in Section 3.1 to highlight the rationale for the data by providing a more detailed explanation and discussion of the data. This study selects two publicly available cybersecurity datasets with industry representativeness: the Canadian Institute for Cybersecurity Intrusion Detection System 2017 (CICIDS 2017) released by the Canadian Institute for Cybersecurity and the Network-Based Intrusion Detection Evaluation Dataset 2015 (UNSW-NB15) developed by the University of New South Wales. The aim is to enhance the model's universality through cross-validation of multi-scenario data features.

5. Additional Suggestions-

Include significance testing (e.g., t-tests or ANOVA) to statistically validate model improvements. Add variability metrics (e.g., standard deviation, confidence intervals) for reported performance scores. Clarify how the attack-aware factor (αc) and skewness threshold (θ) are calculated and adjusted. Provide runtime benchmarks or discuss deployment feasibility for real-time systems. Improve figures for clarity, and ensure all abbreviations are defined on first use. Submit the manuscript for professional language editing.

Reply: Thank you for your suggestions. According to your advice, the content of the results section in this paper has been revised, including adding the results of significance analysis and variability measurement. Then, Section 3.3 clarifies the calculation and adjustment methods of attack perception factors and skewness thresholds. Meanwhile, some figures have been modified to improve the quality. Finally, the manuscript has been professionally edited to enhance the overall content quality. This study focuses on the impact of user group changes on cyberspace vulnerability prediction in the context of capital inflow, proposes a threat-aware support vector machine model, improves the non-linear attack recognition ability through dynamic adaptive kernel optimization, and verifies the effectiveness of the model on the dataset.

Reviewer #2: Thank you to the Author and Editor(s) for this opportunity to provide feedback.

This Original Submission is well-written in both standard and scientific English, well-organized, and its data are largely unproblematic; its original contribution to scholarly research is also evident. I do not have any concerns about its equations or statistical explanations.

Reply: Thank you for your affirmation. Your affirmation is an important supporting factor for the publication of this paper, so I would like to express my heartfelt gratitude to you.

In the real world, organized cyberattacks are rarely categoric or unidimensional, often integrating approaches by disdaining loss, following the path of least resistance and concentrating efforts. I wonder if the Author thinks it is pertinent to include a more detailed comparison of how the AKS mechanism is mobilized against multidimensional attacks, since this would be the best representation of feasibility and the project is designed in-itself for performance.

Reply: Thank you for your suggestions. According to your advice, the content at the end of the results section in this paper has been revised, which discusses how the AKS mechanism mobilizes against multi-dimensional attacks, thereby improving the practicality and value of this study. In real-world multi-dimensional attack scenarios, the AKS mechanism achieves targeted mobilization by dynamically perceiving data distributions and attack features. When it is detected that mixed attacks cause the data skewness to exceed the threshold θ=1.5, the mechanism automatically triggers the Sigmoid kernel to handle sparse boundaries, while adjusting the kernel parameter weights according to the attack perception factor α_c to strengthen the decision boundaries of high-risk attack dimensions.

In real-world applications, the SVM property of “insufficient processing capability of high-dimensional nonlinear data” poses a problem for advanced computation requirements. How does the Author propose to circumvent clear obstacles to implementation? For example, would parameter adjustment, parameter refinement and kernel function switching be feasible simultaneous yet separate processes at larger scales for consumers? I propose the Author include some text about implementation potential, especially since past reviewers seemed primarily skeptical due to an inherent supposition of novel computation leading to restricted implementation.

Reply: Thank you for the questions raised. According to the above questions, the discussion section has been revised to provide a discursive response, thereby enhancing the rationality of this paper and highlighting the implementation potential of this study. In practical applications, aiming at the capability bottleneck of SVM in processing high-dimensional non-linear data, the authors propose a triple optimization mechanism: adopting a lightweight parameter update strategy based on a 500-sample sliding window to reduce the computational load of kernel parameter adjustment; designing a parallel kernel function switching architecture to achieve independent calculation and result fusion of RBF and Sigmoid kernels through CPU multi-threading; introducing a progressive feature dimensionality reduction scheme, which first retains principal components with 95% variance through PCA, and then combines dynamic kernel adjustment. These mechanisms enable the model to maintain high detection accuracy and low memory occupation on edge computing devices, effectively avoiding high-dimensional computational bottlenecks

---

## [Editor Report · Decision Letter 1]

Enhanced SVM-Based Model for Predicting Cyberspace Vulnerabilities: Analyzing the Role of User Group Dynamics and Capital Influx

PONE-D-25-20493R1

Dear Dr. Long,

We’re pleased to inform you that your manuscript has been judged scientifically suitable for publication and will be formally accepted for publication once it meets all outstanding technical requirements.

Kind regards,

Pankaj Bhambri, Ph.D.

Academic Editor

PLOS ONE
---

## [Editor Report · Acceptance letter]

PONE-D-25-20493R1

PLOS ONE

Dear Dr. Long,

I'm pleased to inform you that your manuscript has been deemed suitable for publication in PLOS ONE. Congratulations! Your manuscript is now being handed over to our production team.

Kind regards,

on behalf of

Dr. Pankaj Bhambri

Academic Editor

PLOS ONE